# The Protection and Management of Wapiti in Desert Oases: Bare Land Poses a Limitation to Wapiti Conservation

**DOI:** 10.3390/biology13090737

**Published:** 2024-09-20

**Authors:** Fujie Qiao, Hairong Du, Xia Zhang, Caiping Feng, Zhihua Tan, Yanqin Yu, Zhensheng Liu

**Affiliations:** 1Department of Biological and Food Engineering, Lyuliang University, Lvliang 033001, China; 15765524324@163.com (F.Q.); xiazhang1425@163.com (X.Z.); 19951017@llu.edu.cn (C.F.); 20041024@llu.edu.cn (Z.T.); 19891021@llu.edu.cn (Y.Y.); 2College of Wildlife and Protected Area, Northeast Forestry University, Harbin 150042, China; dhr9012@163.com

**Keywords:** *Cervus canadensis alashanicus*, model parameter optimization, ecological island, habitat assessment, desert oasis

## Abstract

**Simple Summary:**

The Helan Mountains, located at the interface between China’s arid and semi-arid zones, form a natural dividing line in the heart of the desert. Often described as a “desert oasis”, this region functions as an ecological island with a uniquely distinctive geographical setting, making it a focal point for contemporary ecological research. The *Alashan wapiti* (*Cervus canadensis alashanicus*), an isolated population in this area, has become a key conservation concern for the Chinese government. Our multi-scale habitat assessments identified bare land as the principal limiting factor for the availability of suitable habitats for the species. Consequently, the protection and restoration of *Alashan wapiti* habitats are of critical importance, particularly in enhancing the quality of existing habitats. Our findings provide valuable scientific insights to support future conservation and management efforts in the Helan Mountains.

**Abstract:**

The Helan Mountains, situated in the heart of the desert, act as a dividing line between China’s arid and semi-arid zones. Often referred to as a “desert oasis”, they create an ecological island with a uniquely distinctive geographical location, making this area a focal point of contemporary research. Ungulates play a critical role in this ecosystem. The *Alashan wapiti* (*Cervus canadensis alashanicus*), an isolated population of China’s smallest wapiti (*Cervus canadensis*) subspecies, is found exclusively within the Helan Mountains Nature Reserve. The conservation of this isolated population is fraught with challenges, particularly during winter, the harshest season for northern ungulates. Winter habitats are crucial for ensuring population stability. Therefore, we used certain methods, such as factor screening and model parameter optimization to assess habitat suitability using multi-scale species distribution models. The optimized results show that suitable habitats overlap with areas of high vegetation coverage in the Helan Mountains, covering just 588.32 km^2^, which is less than a quarter of the reserve’s total area. The bare land area and winter NDVI are the two primary factors influencing habitat suitability, with other factors having minimal impact, underscoring the critical importance of food resources for the *Alashan wapiti*. The limited availability of these resources poses significant conservation challenges. Our findings provide a more precise foundation for targeted habitat protection and restoration efforts. We recommend enhancing the protection and restoration of food resources, effectively conserving vegetated areas, and preventing desertification.

## 1. Introduction

Large mammalian herbivores play a central role in the structure and functioning of terrestrial ecosystems. Large herbivores consistently exert top-down control over of plant demography, species composition, and biomass, thereby suppressing fires and the abundance of smaller animals (for example, large herbivores suppress pollinators by reducing flower abundance and diversity and rodents by reducing plant density and seed set, which in turn limits predators of these species) [1]. Ungulates play a unique role in the entire ecosystem that is impossible to replace. Wapiti (*Cervus canadensis*) is one of the most widely distributed wild ungulates in the world, and its spatial behavior is highly variable across the species range [2]. The range and habitat selection of wapiti are influenced by a combination of natural and anthropogenic factors. For instance, the extent of a wapiti’s range is determined by certain variables, such as climatic conditions, forage quality and availability, terrain characteristics, population density, predation pressure, and human disturbance [3,4,5]. The *Alashan wapiti* (*C. c. alashanicus*, Figure 1) is one of the eight subspecies of wapiti in China. It is exclusively found in the Helan Mountains and has the smallest distribution range among of all the subspecies [6]. The Helan Mountains are possess a remarkably unique geographical location, being situated in the heart of the desert and forming a distinctive, isolated desert oasis. Despite this, the region experiences limited precipitation, with an annual rainfall of only 200mm, which places it within the arid-semiarid transition zone of northwest China [7]. The low plant diversity results in limited plant resources available to ungulates, making the availability of food resources a critical factor influencing their survival [8].

The most critical factors for the long-term survival of isolated populations are (i) an adequate amount of available habitat and (ii) the genetic diversity within the population [9]. *Alashan wapiti* is threatened by a limited habitat area and low genetic diversity, which are precisely the challenges faced in the conservation of small populations [10]. The *Alashan wapiti* is an isolated, small population, completely separated from other wapiti subspecies [10]. It may experience increased genetic drift and inbreeding, which can lead to a loss of genetic diversity and adaptive flexibility [11], ultimately placing the population at risk of extinction [12,13]. The conservation of the *Alashan wapiti* is of paramount importance within the Helan Mountain ecosystem [14]. Ensuring the scientific protection of its habitat is essential for the long-term survival of the *Alashan wapiti* population [15]. 

A habitat provides a space for a species to live, and its selection is influenced by biotic and abiotic factors [16]. Animals ideally choose habitats that suit them, optimizing access to food, mates, and other resources and thereby enhancing their lifespan and reproductive success [17]. Identifying habitat features is crucial, as habitat loss and degradation are primary drivers of species endangerment, especially when habitats are exposed to climate change and human activities [18]. Habitat selection of animals has been described as scale dependent, with extensive to fine scale selection occurring hierarchically [19,20,21]. Habitat suitability studies often adopt a single-scale approach. However, mounting evidence indicates that biological, ecological, and geographical processes operate at various spatial scales. Additionally, disturbances at one scale can cascade and trigger habitat changes at different scales [22]. Assessing habitat suitability requirements within a multi-scale habitat selection framework allows for a deeper understanding of the key habitat characteristics related to animal adaptability [23]. Many recent studies have observed differences in habitat selection of species across various scales and have emphasized the importance of multiscale comparisons [21,24,25,26,27]. Therefore, multiple scales need to be considered in order to accurately describe the relationship between species and a habitat [28].

Understanding the plasticity of habitat selection across different scales is a necessary condition for implementing effective habitat management actions [29]. Understanding the scale at which the *Alashan wapiti* selects its habitat is a critical factor for ongoing conservation efforts. However, current research on the multi-scale habitat preferences of the *Alashan wapiti* is scarce. Considering the characteristics of ungulates in arid regions, we hypothesize that (1) the habitat selection of the *Alashan wapiti* is influenced by the availability of food resources and that (2) due to resource constraints, the *Alashan wapiti* prefers to select larger-scale habitats. This study aims to offer scientifically-based guidelines for the effective habitat management of ungulates in the Helan Mountains. 

## 2. Method

### 2.1. Study Area 

The Helan Mountains (105°49′~106°42′ E, 38°21′~39°22′ N) extend from the northeast to the southwest, with the highest elevation reaching 3556 meters, creating a vertical difference of 2400 m from the foothills (Figure 2). Consequently, the vegetation exhibits distinct vertical zonation characteristics. As elevation increases, the vegetation transitions sequentially through desert, desert-steppe, steppe, coniferous forest, and shrub meadow. The surrounding plains of the Helan Mountains are desert, encircled by the Ulan Buh Desert, the Tengger Desert, and the Maowusu Desert. This creates what is known as a desert oasis, thus forming an ecological island. Vegetation along the elevation gradient shows a clear vertical zonation: desert steppes are distributed between 2000 and 2300 m, where vegetation is sparse and includes various xerophytic woody plants and shrubs; forests are found between 1800 and 3000 m, with representative plants such as *Picea crassifolia*, *Populus davidiana*, and *Pinus tabuliformis*; and the alpine shrub meadow zone is located around 2800 to 3500 m, where exposed rocks dominate, and vegetation is sparse, with the main species including *Dasiphora parvifolia*, *Pentaphylloides daurica*, and *Kobresia pygmaea*. This region is home to *Pseudois nayaur*, *Moschus chrysogaster*, *Gazella subgutturosa*, *Vulpes vulpes*, *Meles meles*, *Martes foina*, *Lepus capensis*, and *Bos mutus*. Additionally, there are very few individuals of three feline species: *Panthera uncia* (only two individuals, which were released back into the wild after being rescued.), *Prionailurus bengalensis*, and *Otocolobus manul*. The number of predators is low, resulting in unchecked herbivore populations [30]. The sympatric population of *Pseudois nayaur* has increased by 53.17% over the past 15 years, leading to heightened resource competition between the *Alashan wapiti* and other herbivores [31,32,33]. Therefore, the scientific protection of the *Alashan wapiti*’s food resources and habitats is particularly crucial. 

### 2.2. GPS Tracking Records

Data for this study were collected during the winters of 2021, 2022, and 2023, when resources are most scarce. We conducted surveys of the Helan Mountains using a combination of transect and the UAV scanning to investigate wapiti. GPS points of the *Alashan wapiti* distribution were obtained based on observed signs of their presence.

### 2.3. Construction of a Multi-Scale Environmental Covariate Set

Wildlife habitat selection is influenced by various resources or environmental conditions, including food availability, water sources, cover, and levels of disturbance [34]. Based on previous studies, five categories of resources or environmental conditions were selected to study the habitat selection of the *Alashan wapiti*: different vegetation type coverages, topography, food resources, climatic conditions, and human impact (Appendix A). In multi-scale suitable habitats, environmental covariates are combinations of scale information and resource/environmental conditions, thus necessitating the inclusion of additional scale information. The home range of the wapiti was used as the basis for defining spatial scales, with the default home range assumed to be circular. All scales included foraging patches and potential maximum activity ranges [35,36], A series of radii (e.g., 250 m, 500 m, 1000 m, 2000 m, 3000 m, 4000 m, 5000 m, and 6000 m) were set to generate a multi-scale set of environmental covariates. Within this set, the raster layer for a given environmental covariate is a stack of multiple raster layers, with each layer representing a specific scale of the environmental covariate.

### 2.4. Model Selection

Species distribution models (SDMs) are empirical tools that utilize statistically or theoretically derived response surfaces to connect field observations with environmental predictor variables [37]. These environmental predictors, which can exert direct or indirect influences on species, are organized along a gradient from similarity to difference [38]. The MaxEnt model, used for habitat selection modeling, is one of the most advanced and widely adopted tools in the field [39,40,41]. The core principle of the MaxEnt model is to define constraints derived from the environmental covariates at the actual geographic occurrence points of a species. Within these constraints, the model then estimates the potential distribution by maximizing entropy. The resulting probability distribution of species occurrence, under maximum entropy, is deemed the best approximation to of the species’ actual distribution [42]. 

In this study, the MaxEnt model is employed as a multi-scale habitat suitability assessment tool, integrating the occurrence points of the *Alashan wapiti* with various factors potentially affecting its habitat suitability. Habitat suitability is assessed across multiple scales for each factor to ascertain the optimal spatial scale at which the *Alashan wapiti* selects its habitat.

### 2.5. Spatial Filtering

Presence-only datasets often exhibit sampling bias and spatial autocorrelation, which can lead to overfitting and biased model predictions. To mitigate spatial autocorrelation, researchers employ spatial filtering, a method proven to reduce sampling bias and enhance the performance of species distribution models [43]. In this study, a filtering distance of 2 km was used to eliminate spatial autocorrelation, thus ensuring that the minimum distance between presence points is 2 km after filtering. Spatial autocorrelation was removed using the SDM Toolbox v2.5 in ArcGIS software.

### 2.6. Variable Selection

The factor selection process comprised three steps: (1) Univariate selection was employed to determine the optimal scale for variables related to habitat suitability. Given the absence of “absence” data, a presence-only univariate MaxEnt model was utilized to select the variables. To enhance model accuracy, background points were chosen to act as pseudo-absence points, with 2000 random points within the study area selected for this purpose. The MaxEnt model was executed using the “dismo” package in R version 4.2.2. Ten-fold cross-validation was applied, dividing the presence points into ten subsets for model training and validation. The model’s predictive performance was assessed using the average area under the receiver operating characteristic (ROC) curve (AUC). Variables with the highest average AUC values, and AUC > 0.7, were selected for the multi-scale multivariate model (Figure 3). (2) The Correlation between the selected variables can lead to multicollinearity, significantly affecting model performance. Thus, eliminating inter-variable correlation is essential. A Pearson correlation test was conducted to identify and remove correlated variables while retaining those with higher AUC values when |r| > 0.8 (Figure 4). In the end, 9 spatial-scale variables and 5 anthropogenic disturbance variables were retained for subsequent model analysis.

### 2.7. Model Optimization and Selection

Model parameter settings significantly influence model performance and predictive accuracy [44]. The default settings of MaxEnt often result in suboptimal performance [45]. Optimizing the model involves balancing goodness-of-fit with complexity, thereby improving the precision of habitat suitability predictions. The data were randomly divided into 10 subsets, and the model was optimized using regularization multipliers (RM) ranging from 1 to 5 (in increments of 1) and seven different feature class (FC) combinations [L, Q, LQ, LQHPT]; where L = Linear, T = Threshold, Q = Quadratic, P = Product. The optimal model was selected based on the Akaike Information Criterion (AIC), with the model with the smallest AIC value chosen as the best model. This process was conducted using the “ENMeval” package in R version 4.2.2.

### 2.8. Model Validation

Several metrics are used to evaluate species distribution models, including maximum training AUC (AUCTRAIN), the maximum test AUC (AUCTEST), the smallest AUC difference between the training and test data (AUCDIFF), and the Akaike Information Criterion (AIC) [46]. The non-threshold-dependent validation AUC measures the MaxEnt model’s discriminatory power. Higher validation AUC values indicate a stronger ability to differentiate between presence and background points. A validation AUC > 0.9 is considered excellent, 0.8–0.9 is good, 0.7–0.8 is fair, and 0.6–0.7 is poor.

## 3. Results

### 3.1. Determination of the Optimal Scales of Environmental Covariates

We finalized the selection of nine scale factors through a two-step screening process: the percentage of bare area (6000 m, Bare_area), the NDVIwater index (6000 m, NDVIwater), the percentage of deciduous broad-leaved forest (6000 m, DBF), Precipitation of the coldest quarter (5000 m, Bio_19), the topographic wetness index (4000 m, TWI), the mean temperature of the coldest quarter (6000 m, Bio_11), the percentage of crop area (6000 m, Crop), the percentage of shrubland (6000 m, Shrubland), and the mean temperature of the driest quarter (6000 m, Bio_9).

### 3.2. Reslts of Model Optimization and Selection

The model optimization results reveal that the optimal configuration is achieved with a feature type of LQHP and a regularization multiplier of 3, resulting in the smallest AICc value (ΔAIC < 2, Figure 5). Validation of the model demonstrates an AUC value of 0.873, indicating robust performance and high reliability.

### 3.3. Habitat Selection Modeling Based on Environmental Covariates with Optimal Scales

The multi-scale species distribution model fitted for suitable habitats indicates that these habitats align with areas of high vegetation coverage. Using the threshold that maximizes the sum of specificity and sensitivity as the binary cutoff point, the suitable habitat area is 588.32 km^2^, of which 131.46 km^2^ is forest, 207.68 km^2^ is grassland, and other land types occupy relatively smaller areas (Figure 6). The factor contributing most to a suitable habitat is the bare land area, followed by the winter NDVI index, while other factors contribute less (Figure 7).

As the area of bare land increases, habitat quality decreases. When the bare land area is between 0 and 50%, habitat quality declines rapidly, and when it exceeds 50%, the habitat is entirely unused. For the winter NDVI index, habitat suitability increases as the index rises between 1.4 and 7, but beyond 7, habitat suitability levels off (Figure 8).

## 4. Discussion

Amidst the current global background of habitat loss and degradation and declining biodiversity, the necessity of habitat restoration has been firmly established, as emphasized by the United Nations Decade on Ecosystem Restoration (https://www.unenvironment.org/news-and-stories/press-release/new-un-decade-ecosystem-restoration-offers-unparalleled-opportunity, accessed on 1 March 2019). However, the precondition for habitat protection and restoration is a clear understanding of the factors influencing animal habitat preferences, which our research addresses. The Helan Mountains are an ecological island with low resilience to risk. Maintaining biodiversity and restoring habitats are the most effective ways to ensure ecosystem stability.

The *Alashan wapiti* represents the smallest and most isolated population of wapiti in China. By employing a multi-scale approach, we have mapped the suitable habitats for this species, thus addressing recent gaps in habitat assessments. Compared to prior single-scale evaluations, our findings are more precise and optimized. Due to their unique geographical location, the Helan Mountains are a critical biodiversity region in western China, and they have been designated as an important biodiversity conservation area by the Chinese government. As one of the larger ungulates in the region, the *Alashan wapiti* plays an essential role in maintaining ecosystem stability. Currently, the Chinese government has initiated a snow leopard reintroduction plan, with the *Alashan wapiti* becoming one of the prey species for the snow leopard. The success rate of snow leopard reintroduction is closely linked to prey availability. Therefore, effectively increasing the population of *Alashan wapiti* not only helps to maintain vegetation stability but also provides prey resources necessary for the recovery of top predators.

The annual precipitation in the Helan Mountains is only 200 mm, leading to limited plant resources due to the scarcity of water. The scarcity of food and water necessitates that wildlife communities be primarily managed by bottom-up processes, meaning they are limited by resource availability [37]. Clearly, food resources constrain the habitat suitability choices of the *Alashan wapiti*, and our research confirms this from a spatial scale perspective. Large-scale environmental factors influence the habitat selection of the *Alashan wapiti*, as larger spatial scales can provide more resources. This stands in stark contrast to wapiti in resource-rich areas [38]. Multi-level, multi-scale resource selection models have been shown to outperform single-scale, single-level models, resulting in stronger inference and predictive capabilities, and they are the most appropriate for use in conservation planning [47]. These findings provide a reference for spatial conservation planning for the *Alashan wapiti*.

In our study, we found that the main factors influencing habitat suitability are related to food resources, such as NDVI during different seasons and bare land (deserts, bare rocks, etc.). The factor with the greatest contribution to habitat suitability was bare land, followed by winter NDVI. The greater the proportion of bare land, the lower the habitat suitability, indicating that food and water resources are the primary limiting factors for habitat suitability [8]. Our results show that bare land is the most significant factor affecting habitat suitability, suggesting that the *Alashan wapiti* cannot adapt to desert environments. Given that the Helan Mountains are surrounded by deserts, interaction with other subspecies is virtually impossible, posing a challenge for the conservation of this population. NDVI is considered an indicator of food resource abundance, and we selected NDVI from different seasons to reflect seasonal changes in food availability. Our findings show that winter NDVI significantly impacts habitat suitability; the harsh winter conditions in the Helan Mountains, coupled with scarce food resources, limit the availability of edible vegetation for ungulates, highlighting the importance of winter food resources. While anthropogenic disturbances are often a focus in current habitat studies [48,49], our research found that human impact is minimal or even nonexistent (see Figure 6). Human activities around the Helan Mountains are concentrated in the desert areas at the foothills, which the wapiti avoid, and thus, these activities do not impact them.

Overall, our results align with our hypothesis that food resources are the driving factor limiting the availability of suitable habitats for the *Alashan wapiti*. We recommend prioritizing the preservation of habitats with food resources as a vital strategy for the conservation and restoration of this species.

## Figures and Tables

**Figure 1 biology-13-00737-f001:**
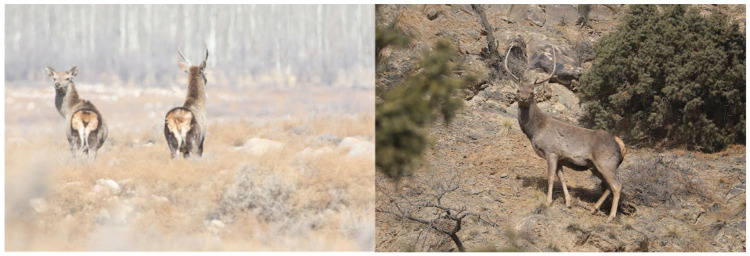
Images of female and male *Alashan wapiti*.

**Figure 2 biology-13-00737-f002:**
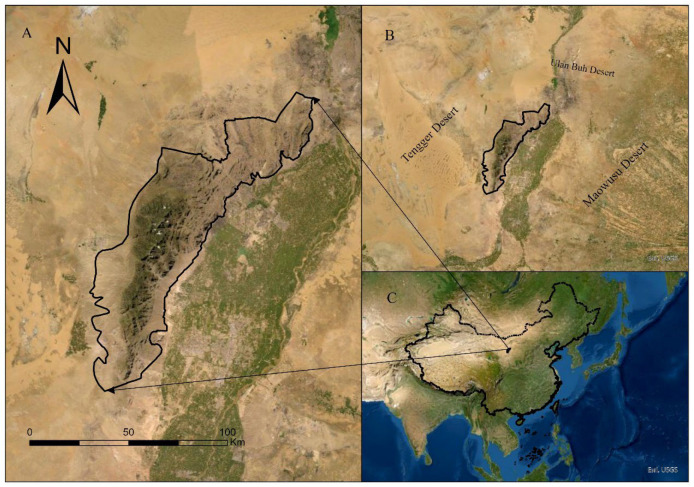
Map of the study area’s geographical location: (**A**): Google Earth image of the study region, showing higher vegetation coverage in the central part of the protected area. (**B**): The protected area is surrounded by the Ulan Buh Desert, the Tengger Desert, and the Maowusu Desert, forming an ecological island. (**C**): The location of the protected area within China.

**Figure 3 biology-13-00737-f003:**
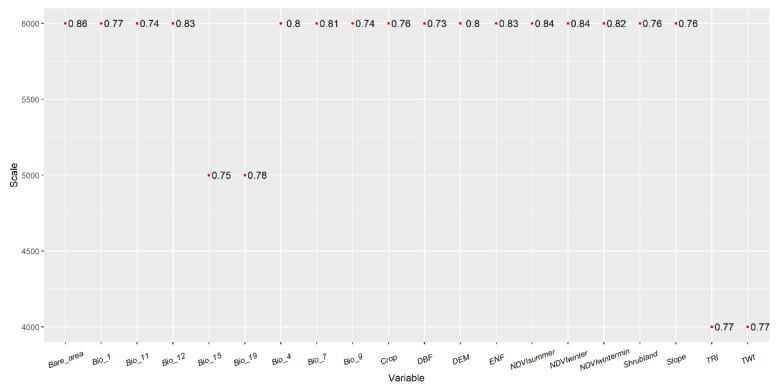
The optimal scale of the variable after AUC screening.

**Figure 4 biology-13-00737-f004:**
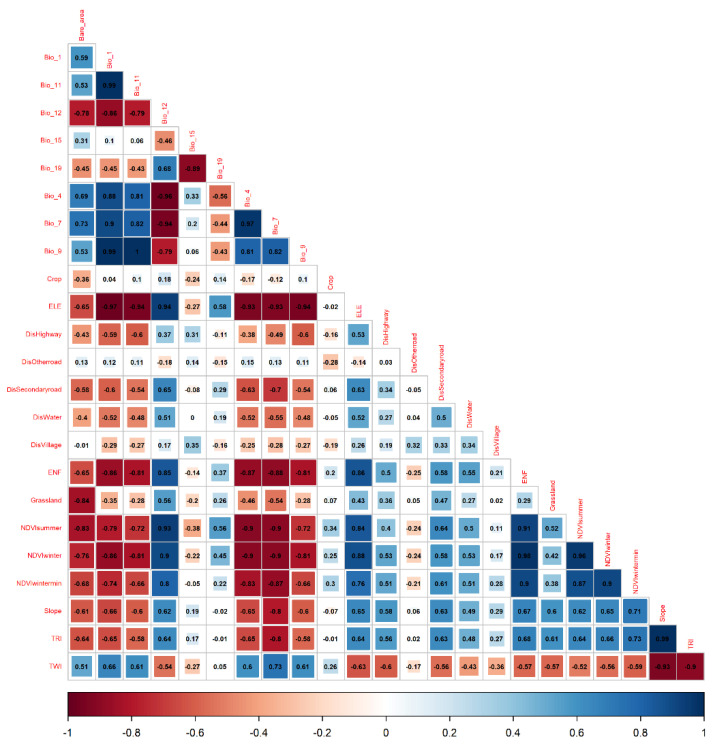
Pearson correlation test results of each variable.

**Figure 5 biology-13-00737-f005:**
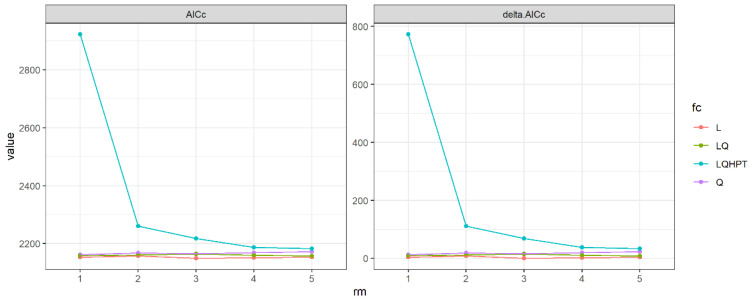
AICc and ΔAICc of the MaxEnt model under different combination of characteristic types and regularized multiplicators.

**Figure 6 biology-13-00737-f006:**
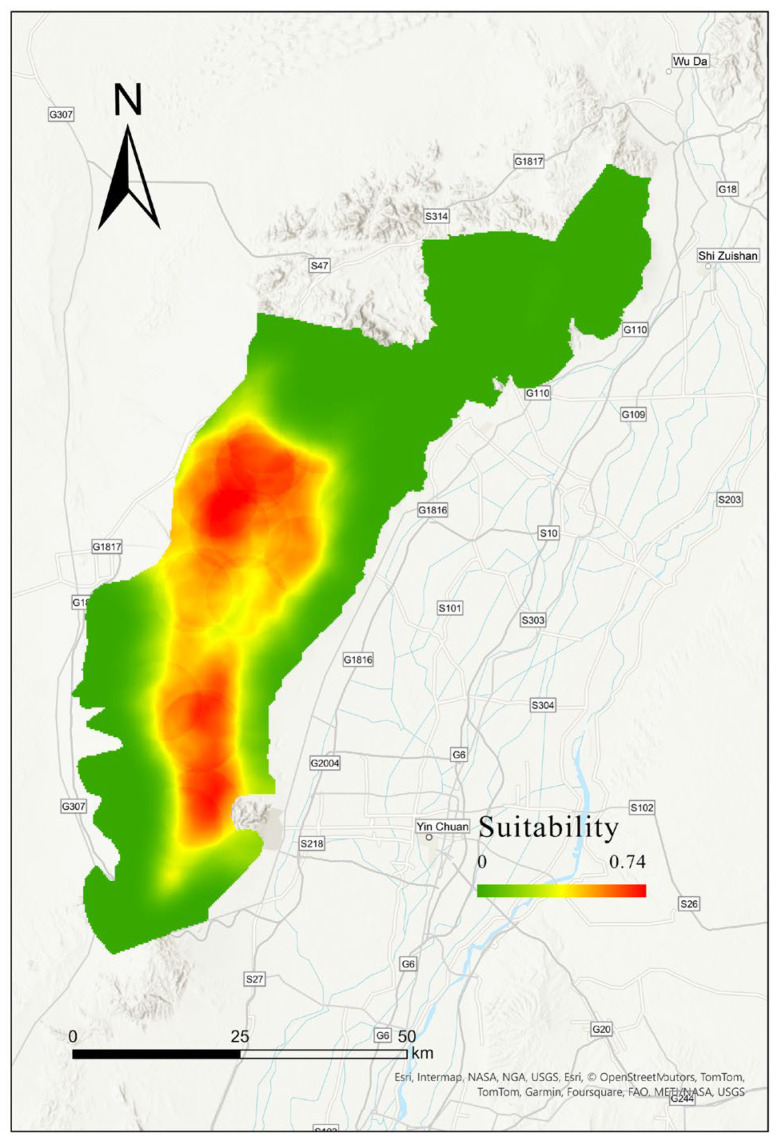
Habitat suitability maps of *Alashan wapiti* based on the optimal scale determined through combinatorial optimization.

**Figure 7 biology-13-00737-f007:**
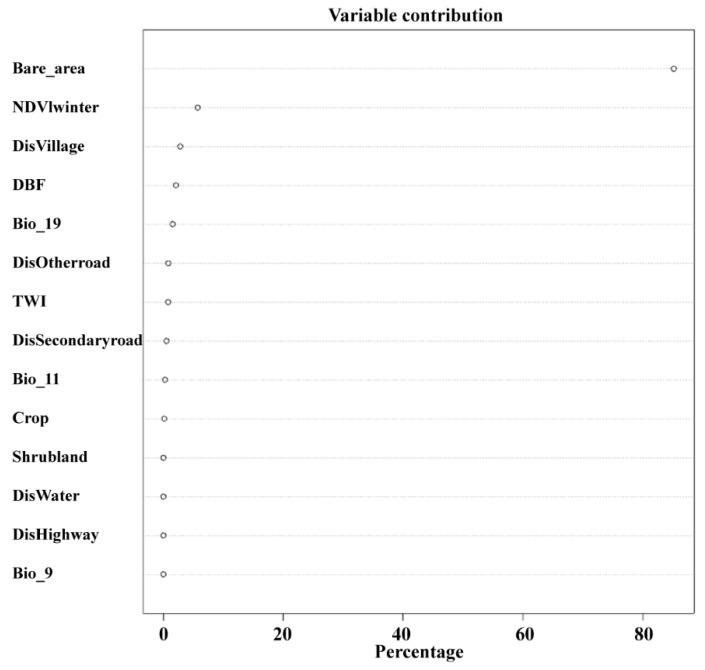
Contribution rates of factors determined through combination optimization for the optimal scale.

**Figure 8 biology-13-00737-f008:**
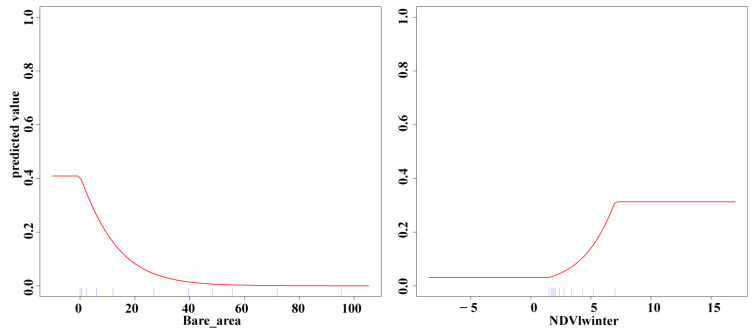
Response curves for the top two environmental covariates in terms of contribution.

## Data Availability

The data used in this study are available from the corresponding author upon request.

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
