# Peer review of "The Protection and Management of Wapiti in Desert Oases: Bare Land Poses a Limitation to Wapiti Conservation"

_biology, 2024, doi:10.3390/biology13090737_

Round 1

Reviewer 1 Report

Comments and Suggestions for Authors

The manuscript describes the use of multi-level, multi-scale resource selection models in obtaining knowledge for spatial conservation planning for the Alashan red deer, the smallest and most isolated population of red deer in China. The data is clearly presented and the study provides new insights that are essential for the Alashan red deer conservation effort.

I recommend unifying the terms used (line 95 - Pseudois nayaur, line 100 - blue sheep). It would be more appropriate to use the scientific name on both lines.

Does the study have any limitations? If so, please add them to the discussion.

Author Response

The manuscript describes the use of multi-level, multi-scale resource selection models in obtaining knowledge for spatial conservation planning for the Alashan red deer, the smallest and most isolated population of red deer in China. The data is clearly presented and the study provides new insights that are essential for the Alashan red deer conservation effort.

1、I recommend unifying the terms used (line 95 - Pseudois nayaur, line 100 - blue sheep). It would be more appropriate to use the scientific name on both lines. Does the study have any limitations? If so, please add them to the discussion.

Response: Thank you very much for your comments, we have made changes to the article.

Reviewer 2 Report

Comments and Suggestions for Authors

I believe the article, "The Protection and Management of Red Deer in Desert Oases: Bare Land Poses a Limitation to Red Deer Conservation," holds significant scientific and practical importance. It highlights the challenges faced by an endangered, isolated population of Ala-Shan deer, a species whose biology remains under-researched (Geist, 1998: Deer of the World). The authors provide an in-depth analysis of the Ala-Shan deer’s habitat and the threats to its preservation. The methodological approach, utilizing a model based on ecological and human disturbance factors, is well-founded. The paper is well-structured, clearly written, with detailed descriptions of the research methods, and the discussion and conclusions are sound and well-supported by the results.

I would, however, like to address the systematic position of the Ala-Shan deer, which the authors have classified as a subspecies of the European red deer (Cervus elaphus). In fact, the Ala-Shan deer is a wapiti, more closely related to Cervus canadensis from Siberia and North America, and represents the subspecies Cervus canadensis alashanicus. I recommend reviewing the following papers, which discuss the taxonomy and evolutionary history of Cervus canadensis in Eurasia:

  • Croitor, R. (2020). A new form of wapiti Cervus canadensis Erxleben, 1777 (Cervidae, Mammalia) from the Late Pleistocene of France. Palaeoworld, 29(4), pp.789-806.
  • Croitor, R., & Obada, T. (2018). On the presence of late Pleistocene wapiti, Cervus canadensis Erxleben, 1777 (Cervidae, Mammalia) in the Palaeolithic site Climăuți II (Moldova). Contributions to Zoology, 87(1), pp.1-10.
  • Geist, V. (1998). Deer of the World: Their Evolution, Behavior, and Ecology. Mechanicsburg: Stackpole Books.

While it is not obligatory to include these sources in your paper's bibliography, I suggest reviewing them to gain a deeper understanding of the taxonomic distinction between Cervus elaphus and Cervus canadensis, which are vicarious species.

I also have a few minor suggestions for improvement:

  • Page 6, line 197: There appears to be an unintended paragraph break here.
  • Figure 4 is missing a number and an explanatory caption.
  • General comment on figures: The axis labels in the plots are too small and difficult to read. Please increase the font size.
  • Page 9, line 249: Consider changing "apex predators" to "top predators" for clarity.

Once these minor revisions are addressed, I would be pleased to recommend the article for publication.

Author Response

I believe the article, "The Protection and Management of Red Deer in Desert Oases: Bare Land Poses a Limitation to Red Deer Conservation," holds significant scientific and practical importance. It highlights the challenges faced by an endangered, isolated population of Ala-Shan deer, a species whose biology remains under-researched (Geist, 1998: Deer of the World). The authors provide an in-depth analysis of the Ala-Shan deer’s habitat and the threats to its preservation. The methodological approach, utilizing a model based on ecological and human disturbance factors, is well-founded. The paper is well-structured, clearly written, with detailed descriptions of the research methods, and the discussion and conclusions are sound and well-supported by the results.

I would, however, like to address the systematic position of the Ala-Shan deer, which the authors have classified as a subspecies of the European red deer (Cervus elaphus). In fact, the Ala-Shan deer is a wapiti, more closely related to Cervus canadensis from Siberia and North America, and represents the subspecies Cervus canadensis alashanicus. I recommend reviewing the following papers, which discuss the taxonomy and evolutionary history of Cervus canadensis in Eurasia:

Croitor, R. (2020). A new form of wapiti Cervus canadensis Erxleben, 1777 (Cervidae, Mammalia) from the Late Pleistocene of France. Palaeoworld, 29(4), pp.789-806.

Croitor, R., & Obada, T. (2018). On the presence of late Pleistocene wapiti, Cervus canadensis Erxleben, 1777 (Cervidae, Mammalia) in the Palaeolithic site Climăuți II (Moldova). Contributions to Zoology, 87(1), pp.1-10.

Geist, V. (1998). Deer of the World: Their Evolution, Behavior, and Ecology. Mechanicsburg: Stackpole Books.

While it is not obligatory to include these sources in your paper's bibliography, I suggest reviewing them to gain a deeper understanding of the taxonomic distinction between Cervus elaphus and Cervus canadensis, which are vicarious species.

Response: We have changed "red deer" to "wapiti"

I also have a few minor suggestions for improvement:

Page 6, line 197: There appears to be an unintended paragraph break here.

Response: Thank you very much for your suggestion, this is a mistake, we have made a modification.

Figure 4 is missing a number and an explanatory caption.

Response: Thank you very much for your suggestion, we have added it “AICc and ΔAICc of Maxent model under different combination of characteristic types and regularized multiplicators”

General comment on figures: The axis labels in the plots are too small and difficult to read. Please increase the font size.

Response: Thank you very much for your comments, we have made changes to the article.

Page 9, line 249: Consider changing "apex predators" to "top predators" for clarity.

Response: Thank you very much for your suggestion,We have changed " apex predators " to " top predators ".

Once these minor revisions are addressed, I would be pleased to recommend the article for publication.

Reviewer 3 Report

Comments and Suggestions for Authors

Premise

Chinese zoologists often continue to consider the Cervus-elaphus-complex a monophyletic large group of subspecies occurring from Europe to N America, despite the fact that genetic studies have confirmed since the late Nineties (Polziehn & Strobeck 1998, Kuwayama & Ozawa 2000, Randi et al. 2001, Ludt et al. 2004, Lorenzini et al. 2005, Heckeberg 2020, Mackiewicz et al. 2022) the need to split the superspecies in (two/)three distinct species (Western red deer C. elaphus of Europe and N Africa, Central Asian C. hanglu of Turkestan, Tarim Basin and Kashmir including yarkandensis, and wapiti C. canadensis of China, Siberia and N America including the subspecies wallichii, kansuensis, mcneilli, alashanicus, xanthopygus, songaricus, sibiricus and the American canadensis). Also in the IUCN Red List the old superspecies is now divided into three species (Lovari et al. 2018 for C. elaphus), Brook et al. 2017 for C. hanglu and Brook et al. 2018 for C. canadensis). Some Chinese zoologists continue to name the Alashan or Helan wapiti as alxaicus instead of alashanicus (see Zhang et al. 2007, Li et al. 2022, Gao et al. 2023), a name coined by Bobrinskii and Flerov in 1935, but this is not the case of the authors of this manuscript, which adopted the correct alashanicus. Recently Xiao et al. (2022) classified alashanicus as a subspecies of C. canadensis and Chinese zooarchaeologists (Song, Zhang, Bao, Cai 2024 “Ancient DNA study of Cervucanadensis unearthed from the Royal Sacrificial Site of the Northern Wei Dynasty in Inner Mongolia Autonomous Region, China”,  Journal of Archaeological Science: reports 57:  104633) adopted canadensis for the Helan wapiti.

Anyway, if the authors prefer to maintain the traditional taxonomy they should at least mention in this ms the alternative division into three species proposed decades ago.

Given the fact that images of alashanicus are extremely rare (actually only a simple sketch in the book “Deer of the World” by V. Geist), it would be interesting to add a photo of it, possibly of an adult male with antlers.

The paper is worth publishing after some implementation, especially in the introduction. You should better explain the taxonomic position of alashanicus, which they continue to include in the old superspecies Cervus elaphus and that in many modern works is a subspecies of Cervus canadensis as Alashan or Helan wapiti. It would be important to add also a short description of the physical characteristics of  this subspecies and mention the last population estimated numbers. In the paragraph on the study area something should be added about the main plant associations. In the Results you should clarify the extension of different main habitats within the most suitable range.

I add some notes or suggestions on specific points:

Lines 34-35: suppressing the abundance of smaller animals?? Not clear

Line 41: Is Alashan another name for Helan?

41: human occurrence? Human disturbance?

54: isolated    repetition (see line 52)

54: How small is this population? How many deer are estimated occurring in Helan Mountains? They were 1700 in 2005 (Zhang et al. 2007); and now?

55: [10].

54-56: Actually it would be very useful here a short description of the characteristics of this subspecies (mean body size of adult males and females, features of the antlers, colour patterns of the rump patch) compared with those of the subspecies occurring nearby (xanthopygus, mcneilli).

As stated above it would be advisable to insert a photo of the deer, especially an adult male with antlers.

91-93: In a paper on habitat suitability you should shortly describe the main plant associations of steppes, forests and shrub meadows. Please implement the description of habitats (as in Zhang et al. 2007, Liu et al. 2020).

Line 213: Please specify the division of the 588 km2 suitable range into main habitats. How much of the range is represented by steppes, forests etc?  How much of the range is over 2000 m asl?

Lines 308, 313: Please quote in full only the surname of the authors

313, 318 (and 334, 335, 357, 375): Names of genus with a capital letter, names of species and subspecies all in lowercase.

313-314: Other similar works Li et al. 2022 “Coexistence mechanisms of sympatric ungulates…”, Frontiers in Ecology and Evolution 10: 925465 and Liu et al. 2018 “Comparative analysis of winter diets and habitat use…” Folia Zoologica 67: 43-53. Please add them.

Please quote somewhere in the text Gao et al. 2023 “Inferring landscape factors driving microgeographic genetic structure of large-sized mountain ungulates: A case of Alashan red deer (Cervus elaphus alxaicus)”, Global Ecology and Conservation 44: e02497 which provides interesting insights on the subspecies.

Comments on the Quality of English Language

The English needs minor editing

Author Response

Chinese zoologists often continue to consider the Cervus-elaphus-complex a monophyletic large group of subspecies occurring from Europe to N America, despite the fact that genetic studies have confirmed since the late Nineties (Polziehn & Strobeck 1998, Kuwayama & Ozawa 2000, Randi et al. 2001, Ludt et al. 2004, Lorenzini et al. 2005, Heckeberg 2020, Mackiewicz et al. 2022) the need to split the superspecies in (two/)three distinct species (Western red deer C. elaphus of Europe and N Africa, Central Asian C. hanglu of Turkestan, Tarim Basin and Kashmir including yarkandensis, and wapiti C. canadensis of China, Siberia and N America including the subspecies wallichii, kansuensis, mcneilli, alashanicus, xanthopygus, songaricus, sibiricus and the American canadensis). Also in the IUCN Red List the old superspecies is now divided into three species (Lovari et al. 2018 for C. elaphus), Brook et al. 2017 for C. hanglu and Brook et al. 2018 for C. canadensis). Some Chinese zoologists continue to name the Alashan or Helan wapiti as alxaicus instead of alashanicus (see Zhang et al. 2007, Li et al. 2022, Gao et al. 2023), a name coined by Bobrinskii and Flerov in 1935, but this is not the case of the authors of this manuscript, which adopted the correct alashanicus. Recently Xiao et al. (2022) classified alashanicus as a subspecies of C. canadensis and Chinese zooarchaeologists (Song, Zhang, Bao, Cai 2024 “Ancient DNA study of Cervus canadensis unearthed from the Royal Sacrificial Site of the Northern Wei Dynasty in Inner Mongolia Autonomous Region, China”,  Journal of Archaeological Science: reports 57:  104633) adopted canadensis for the Helan wapiti.

Anyway, if the authors prefer to maintain the traditional taxonomy they should at least mention in this ms the alternative division into three species proposed decades ago.

Given the fact that images of alashanicus are extremely rare (actually only a simple sketch in the book “Deer of the World” by V. Geist), it would be interesting to add a photo of it, possibly of an adult male with antlers.

The paper is worth publishing after some implementation, especially in the introduction. You should better explain the taxonomic position of alashanicus, which they continue to include in the old superspecies Cervus elaphus and that in many modern works is a subspecies of Cervus canadensis as Alashan or Helan wapiti. It would be important to add also a short description of the physical characteristics of this subspecies and mention the last population estimated numbers. In the paragraph on the study area something should be added about the main plant associations. In the Results you should clarify the extension of different main habitats within the most suitable range.

Response: Your suggestions are excellent, and we are very grateful. Your feedback has been very helpful to us, and we have already made the corresponding changes in the relevant sections of the article."

I add some notes or suggestions on specific points:

Lines 34-35: suppressing the abundance of smaller animals?? Not clear

Response: We have added to the article :“For example, large herbivores suppress pollinators by reducing flower abundance and diversity and rodents by reducing plant density and seed set, which in turn limits predators of these species”

.Line 41: Is Alashan another name for Helan?

Response: Alashan is the name of a "city". China named the wapiti in Helan Mountain as Alashan wapiti.

41: human occurrence? Human disturbance?

Response: We have already changed 'human development' to 'human disturbance'.

54: isolated    repetition (see line 52)

Response: We have already made revisions to the repetitive sentences.

54: How small is this population? How many deer are estimated occurring in Helan Mountains? They were 1700 in 2005 (Zhang et al. 2007); and now?

Response: A transect study found that the population of wapiti in the winter of 2018 reached a peak of approximately 2,452 individuals (1,678- 3,578).

54-56: Actually it would be very useful here a short description of the characteristics of this subspecies (mean body size of adult males and females, features of the antlers, colour patterns of the rump patch) compared with those of the subspecies occurring nearby (xanthopygus, mcneilli).

As stated above it would be advisable to insert a photo of the deer, especially an adult male with antlers.

Response: We have already added the photos to the article.

91-93: In a paper on habitat suitability you should shortly describe the main plant associations of steppes, forests and shrub meadows. Please implement the description of habitats (as in Zhang et al. 2007, Liu et al. 2020).

Response:  We have already added the corresponding content to the article. “Vegetation along the elevation gradient shows a clear vertical zonation: desert steppes are distributed between 2,000 and 2,300 meters, where vegetation is sparse and includes various xerophytic woody plants and shrubs; forests are found between 1,800 and 3,000 meters, with representative plants such as Picea crassifolia, Populus davidiana, and Pinus tabuliformis; the alpine shrub meadow zone is located around 2,800 to 3,500 meters, where exposed rocks dominate, and vegetation is sparse, with main species including Dasiphora parvifolia, Pentaphylloides daurica, and Kobresia pygmaea.”

Line 213: Please specify the division of the 588 km2 suitable range into main habitats. How much of the range is represented by steppes, forests etc? 

Response: steppes:207.68km2; forests:131.46km2; Other land types are very small and we did not count them.

Lines 308, 313: Please quote in full only the surname of the authors

Response: Thank you for your comments, we have made the changes.” Luo Y. Comparing the Diet and Habitat Selection of Sympatric Blue Sheep (Pseudois nayaur) and Reddeer (Cervus elaphus alxaicus) in Helan Mountains, China. D, Northeast Forestry University, 2011.”

313, 318 (and 334, 335, 357, 375): Names of genus with a capital letter, names of species and subspecies all in lowercase.

Response: Thank you for your suggestion, we have made the changes.

313-314: Other similar works Li et al. 2022 “Coexistence mechanisms of sympatric ungulates…”, Frontiers in Ecology and Evolution 10: 925465 and Liu et al. 2018 “Comparative analysis of winter diets and habitat use…” Folia Zoologica 67: 43-53. Please add them.

Response: Thank you for your suggestion, we have made the changes.

Please quote somewhere in the text Gao et al. 2023 “Inferring landscape factors driving microgeographic genetic structure of large-sized mountain ungulates: A case of Alashan red deer (Cervus elaphus alxaicus)”, Global Ecology and Conservation 44: e02497 which provides interesting insights on the subspecies.

Response: Thank you for your suggestion, We have already cited the article you recommended.